# Long-term conservation agriculture with optimum nitrogen fertilization improves soil phosphorus availability

Nusrat Jahan Mumu[1,2]*, Sunjana Akter[2], Afsana Mimi Eiti Mony[2], Jannatul Ferdous[2], Nushaiba Atiq Taima[2], Most. Khatiza Khatun[2], Md. Mofizur Rahman Jahangir[2]

1 Department of Soil Science, Khulna Agricultural University, Khulna, Bangladesh, 2 Department of Soil Science, Bangladesh Agricultural University, Mymensingh, Bangladesh

* nusratmumu43044@bau.edu.bd

## Abstract

Understanding the interactive effects of conservation agriculture (CA) and nitrogen (N) fertilization on soil phosphorus (P) dynamics is critical for sustainable nutrient management. However, information on the dynamics and availability of P pools under the combination of long-term CA practices with N fertilization is limited. This study aimed to evaluate the long-term impacts of CA and different N rates on labile, moderately labile, and non-labile P fractions in a wheat (*Triticum aestivum*)-mungbean (*Vigna radiata*)-rice (*Oryza sativa* L) rotation after the 36th consecutive crop. The field experiment was conducted in a split-split plot design with two tillage systems (conventional tillage, CT; strip tillage, ST), two residue levels (lower residue, LR (15 cm); higher residue, HR (30 cm)), and six nitrogen (N) rates (N0, N60, N80, N100, N120, N140 representing 0%, 60%, 80%,100%, 120% and 140%, respectively of the recommended dose). The ST-HR significantly enhanced labile-P availability. Soluble-P, and $NaHCO_3$-Po (organic) increased by 4% and 35% under ST compared to CT, and by 259% and 26% under HR compared to LR, respectively. While HR enhanced most soil P fractions, higher N rates (>N100) tended to decrease labile-Pi (inorganic) fractions by up to 45% suggesting a potential decline in plant-available P. The $NaHCO_3$-Pi was higher in HR coupled with lower N rate (≤N100). Optimum N rate under CA aided in raising the $NaHCO_3$-Po fractions. Moderately labile-P fractions were higher in ST-HR with increased N rate (>N100). Acid-P increased by 38% under ST, with complex interactions observed across treatments. Acid-P was consistently higher in ST-HR coupled with lower N rate (≤N100) while residual-P was higher in ST-LR coupled with same N rate. The interactions among tillage, residue, and N were significant for most P pools highlighting the synergistic effects of CA and N management. Overall, ST-HR-N100 was found to optimize P availability and minimize non-labile P buildup, offering a balanced P dynamics for improved soil fertility and sustainable crop production.

**Data availability statement:** All relevant data are within the paper and its Supporting information files.

**Funding:** This research work was financially supported by the Krishi Gobeshona Foundation, Bangladesh (LWR/2016/136: KGF (tbc); to Md. Mofizur Rahman Jahangir).

**Competing interests:** The authors have declared that no competing interests exist.

## Introduction

Soil phosphorus (P) availability is highly sensitive to long-term agronomic practices. Conventional intensive cropping practices, including continuous tillage, monocropping, reliance on chemical fertilizers, and inadequate residue management often accelerate P fixation and deplete labile P pools, leading to declining soil fertility [1,2]. In contrast, conservation agriculture (CA) practices which include minimum soil disturbance, crop residue addition, and rotation with diversified crops can contribute to enhanced P cycling and availability [3,4]. Crop residues play a dual role; they contribute to organic P (Po) pools through organic matter inputs and influence P mineralization depending on their C:P ratios and decomposition dynamics [5,6]. In particular, conservation tillage and crop residue retention have shown potential to increase the availability of labile P fractions by reducing soil erosion and P fixation [4–6].

In agricultural soils, P exists in two broad forms: organic P (Po) and inorganic P (Pi), each of which can be further divided into fractions based on their availability-labile (solution P, $NaHCO_3$-Pi, and $NaHCO_3$-Po), moderately-labile (NaOH-Pi, NaOH-Po), and non-labile (acid P and residue P) P fractions [7–9]. The equilibrium between these fractions, especially between labile and moderately labile P, governs the short-term dynamics of P supply to crops [10,11]. Plant-available P is largely confined to the labile pool, which is typically small due to the strong tendency of P to bind with soil particles, resulting in fixation [8,11], and over 90% of total soil P is typically found in insoluble and fixed forms [12], limiting its availability and thus crop productivity. Moreover, nitrogen (N) fertilization can mediate the P cycle by affecting microbial-mediated processes, soil pH, and organic matter turnover [2]. Excessive application of ammonium-based fertilizers like urea can acidify the soil, thereby increasing P sorption and reducing labile Pi availability [13]. However, when optimum N is applied under CA conditions, it can stimulate microbial activity and enhance the mineralization of Po, thus increasing Pi availability [2,14]. This N–P interaction is highly context-specific, depending on soil type, cropping system, and climatic conditions. Therefore, location and crop-specific assessments of P behavior in response to N fertilization is crucial for balanced nutrient management and crop production.

Previous research has identified that soils in the Old Brahmaputra Floodplain (AEZ-9) contain higher levels of Po than Pi, making organic P management critical for sustained productivity [15]. Long-term cereal cultivation under varying tillage, residue incorporation and fertilization regimes may lead to significant changes in distribution and availability of different soil P fractions over time in this region. Conservation agriculture practices especially surface residue retention and reduced tillage promote the stratification of P in the upper soil layers, where residues decomposition and root density are highest [16–19]. This stratification has important implications on nutrient uptake, particularly in rainfed or shallow-rooted cropping systems where surface Pi is more accessible, but may limit deeper root access under drought or moisture-stress conditions [20]. Soil P stratification has some other environmental and agronomic implications such as loss of P through runoff and leaching and eutrophication of lake [21,22]. Thus, understanding the extent and functional implications of P stratification is essential for optimizing P use efficiency and crop productivity under CA. Despite

its importance, there remains a lack of data and the extent of P availability in crop root zone under CA and how it interacts with N fertilization to influence the distribution and transformation of P fractions over time. Addressing these knowledge gaps is essential for designing nutrient management strategies that utilize CA, optimize P availability, minimize fixation, and sustain productivity under long-term cereal-based cropping system in tropical areas.

Most previous studies have focused either on total P content or crop response, without detailing the shifts among P fractions that determine the availability. Older fractionation methods such as those developed by Chang and Jackson [23], Hingston et al. [24] have been largely replaced by modified sequential extraction techniques [25,26], which allow for a more precise understanding of labile, moderately labile, and non-labile P pools [27]. Maniruzzaman et al. [28] used modified Hedley procedure [25] to determine soil P fractionations in calcareous soils and found that short-term CA practices, particularly increased crop residue retention, significantly elevated almost all P fractions and enhance overall P stocks in topsoil. Reduced tillage was found having smaller yet notable effects by increasing inorganic P fractions in upper layer of soils highlighting residue retention as the dominant driver of P fraction shifts [28]. Our research intends to extend these findings through long-term CA practices and N fertilizer application in a field experiment, specifically within South Asian cereal-based systems, aiming to clarify how these practices influence the turnover and availability of distinct P pools over time.

Moreover, the synergistic effects of CA components such as minimal soil disturbance and residue retention can improve soil physical properties, increase organic matter inputs, and promote microbial activity [29–31]. These factors, in turn, stimulate the mineralization of Po and reduce P adsorption by stabilizing soil aggregates and modifying the chemical environment of the rhizosphere [32–34]. Retaining crop residues in combination with reduced tillage can lessen soil P fixation, increase labile P content, and promote the development of Po and phosphatase-mediated mineralization [35]. The conversion of labile P forms into moderately available and non-labile forms can be facilitated by N fertilization through various soil mechanisms [14]). Balanced N fertilization can enhance microbial biomass and enzymatic activity, thereby facilitating the conversion of Po to plant-available Pi forms. However, excessive or imbalanced N inputs can acidify the soil and increase P sorption, converts labile Pi into recalcitrant P or moderately accessible P [36,37]. The present study hypothesizes that CA combined with optimal N fertilization will enhance soil P availability for plant uptake by increasing labile and organic P fractions while minimizing P fixation. In addition, the positive interaction of CA with N inputs can be leveraged for sustainable nutrient management influencing the transformation and mobilization of soil P. The objectives of this study were (i) to assess the long-term effects of CA with varying N fertilization on the dynamics and distribution of soil P fractions in a cereal-based cropping system; and ii) to identify the optimum N fertilization rate that minimizes P fixation and enhances plant-available P fractions under CA practices.

## Materials and methods

### Site description

The research was conducted on the Soil Science Field Laboratory (24°71.60'N, 90°42.51'E) at Bangladesh Agricultural University, Mymensingh in Bangladesh. The field site is located on the Old Brahmaputra Floodplain soil, and is classified as non-calcareous dark grey floodplain soil under the Sonatala series. It has a silty loam texture, is neutral in pH (6.5), and poorly drained. With an average yearly temperature of 26° C, 1800 mm of rainfall, and 85% relative humidity, the area experiences sub-tropical monsoon weather (Weather station, BAU). Farmers in Mymensingh region commonly follow the wheat-fallow-rice cropping pattern but, in the experiment, we included mungbean which is a short duration leguminous crop, as a CA practice. Thus, the experiment followed a wheat (BARI hybrid maize-14) – mungbean (BARI Mung-6) –rice (BRRI-71) cropping pattern which was continued at this location for 12 years.

### Experimental design and treatment applications

In the experimental site, CA (minimum tillage, soil cover residue, and adding short-duration legumes to crop rotation) was implemented as a long-term practice since 2011. There were three sets of treatments in the split-split plot

design of the experiment: 1) conventional tillage (CT) and strip tillage (ST- only the rows (2–3 cm) were tilled and the areas (20 cm) in between the rows were left untilled, tilled rows were used for planting crops) were assigned to the main plots; 2) crop residue retention was assigned to the sub-plots (higher residue, HR: 30 cm of rice and wheat residue + 100% mungbean residue, by height; lower residue, LR: 15 cm of rice and wheat residue + 100% mungbean residue); and 3) Nitrogen fertilizer dose: N0: 0% of RD (recommended dose); N60: 60% of RD; N80: 80% of RD; N100: 100% of RD; N120: 120% of RD; N140: 140% of RD were assigned to sub-sub plots with three replications. Recommended fertilizer dose was followed from the national fertilizer recommendation guide for the location-specific test crops [38]. The 100% RD for cultivating wheat, mung bean and rice at the experimental location was 100 kg N/ha, 20 kg N/ha and 120 kg N/ha, respectively.

Each sub-sub plot was 13 m × 3 m and had a dividing bund around each plot. Urea, triple super phosphate, muriate of potash, gypsum, magnesium sulphate, zinc sulphate and boric acid were used for wheat, mungbean and rice as sources of N, P, K, S, Mg, Zn and B, respectively and the fertilizers were applied following the national fertilizer recommendation dose [38] for this region known as Agro-ecological Zone (AEZ) 9. During the final land preparation, a basal dose of every fertilizer had been put to every plot, with the exception of urea during rice and wheat cultivation. Urea was applied to each plot in three equal split applications in wheat and rice. Moreover, intercultural practices including irrigation, weeding, insect and disease management were done aligned with the cultural practices for Bangladesh AEZ-9.

## Soil sample collection, processing and analysis

Composite soil samples were collected from each plot in 2023 after thirty-six consecutive crops (12 years following the repeated cropping sequence and CA). Soil samples were collected after crop harvesting using an auger from five different spots per plot randomly at 0–15 cm soil depth. The collected soil samples were mixed thoroughly to make a composite soil sample, air-dried at room temperature, passed through a 2 mm sieve, and stored until further analysis. One gram of finely grounded soil sample (0.2 mm) was extracted sequentially for soil P fractionations (Fig 1) following the method described by Hedley [26], with a further modification involving the final soil digestion by nitric acid and perchloric acid [39]. The sequential extractions of seven P fractions including solution P, $NaHCO_3$-P (Pi and Po), NaOH-P (Pi and Po), acid P and residual P were done (Fig 2).

Solution P was extracted by 0.01 M $CaCl_2$ for inorganic P fraction. Soil $NaHCO_3$-P fraction was extracted by 0.5 M $NaHCO_3$ and $NaHCO_3$-Po (organic) were estimated by deducting $NaHCO_3$-Pi (inorganic) from the total $NaHCO_3$-extractable P following $H_2SO_4$ and $H_2O_2$ digestion. NaOH-P fraction was extracted by 0.1 M NaOH. After being digested with $H_2SO_4$ and $H_2O_2$, NaOH-Pi was deducted from the total NaOH-extractable P to determine NaOH-Po. Acid-P was extracted by 0.5 M $H_2SO_4$ and residual-P was extracted by $HNO_3$ and $HClO_4$ (5:2 ratio). The concentration of P in the extracts and digests was measured using the ammonium molybdate-ascorbic acid method after neutralization (if necessary) and the absorbance of P was determined using a spectrophotometer at 882 nm wavelength [40].

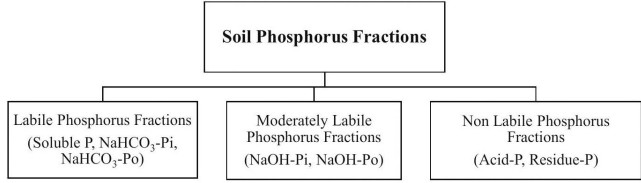

**Fig 1. Different soil phosphorus fractions according to lability.**

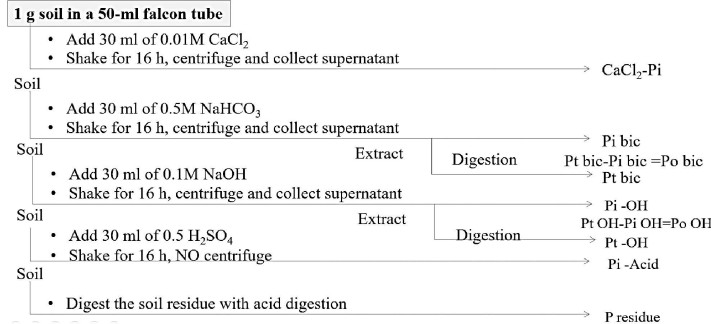

**Fig 2. Soil phosphorus fractionations procedure [41].**

## Statistical analysis

Tillage, residues, and N rate were the fixed factors in a split-split plot (three-way analysis) analysis of variance (ANOVA). To determine the impact and interactions between and among tillage, residue, and N rate treatments, data was statistically analysed using the Jamovi 1.0.0.0. (R Package) software. The comparison of means was tested by the Duncan's Multiple Range Test (DMRT) and least significant difference (LSD) at the $p < 0.05$ level using SPSS (IBM SPSS Statistics, version 20, IBM Corp).

## Results

### Effects of management practices on labile phosphorus fractions

Soluble-P content was significantly influenced by the interaction of tillage, residue retention, and nitrogen (N) application rate (P < 0.001, Table 1). The highest soluble-P (13.4 mg kg$^{-1}$) was observed under CT-HR-N100, which was over 12 times higher than the lowest value (1.0 mg kg$^{-1}$) in CT-LR-N120. Soluble-P decreased with increasing N rates beyond N100, particularly under low residue (LR) treatments in either tillage.

The interaction effects of tillage, residue, and N levels were significant for the NaHCO$_3$-inorganic P (NaHCO$_3$-Pi) fraction (P < 0.001, Table 1), ranging from 2.4 to 46.0 mg kg$^{-1}$. The highest value (46.0 mg kg$^{-1}$) was recorded under ST-HR-N0, which was about 90% higher than ST-HR-N100 (13.2 mg kg$^{-1}$). Notably, NaHCO$_3$-Pi decreased sharply with increasing N rates under both ST-HR and ST-LR (Table 1). Combination of CT-HR with lower N rates considerably raised the soils' NaHCO$_3$-Pi fraction compared to CT-LR with similar N rates. Likewise, the NaHCO$_3$-organic P fractions (NaHCO$_3$-Po) varied significantly among the treatment combinations (P < 0.001, Table 1). The ST-HR-N100 treatment showed the highest NaHCO$_3$-Po (36.0 mg kg$^{-1}$), about 5 times increase over the minimum value observed under ST-LR-N140 (7.8 mg kg$^{-1}$) treatment. Higher N rates (> N3) tended to decrease NaHCO$_3$-Po levels across tillage and residue treatments.

### Effects of management practices on moderately labile phosphorus fractions

Tillage, residue, and N rates significantly influenced NaOH-extractable inorganic P (NaOH-Pi) fractions (P < 0.001, Table 2). The NaOH-Pi ranged from 5.6 to 38.2 mg kg$^{-1}$, with the highest value recorded in the ST-HR-N140 treatment, nearly 7-fold higher than the minimum in CT-HR-N0 treatment combination. The NaOH-Pi fraction was significantly increased with elevated N rates combining with CT-HR, ST-HR and ST-LR treatments. Notably, NaOH-Pi levels in ST consistently outperformed CT under similar residue and N treatments.

No significant interaction among tillage, residue and N rates was found for NaOH-Po (p > 0.05, Table 2) fractions. The soil NaOH-Po fractions ranged from 7.4 to 54.6 mg kg$^{-1}$ across all treatments. Mean NaOH-Po concentrations under ST were 60% higher than under CT, and HR showed nearly double NaOH-Po concentrations compared to LR treatments.

**Table 1. Labile-P fractions in soil (depth: 0-15 cm) under long term conventional (CT) and strip (ST) tillage systems; high residue (HR) and low residue (LR) retention; and six N application rates.**

| Treatments | | | Labile Phosphorus Fractions | | |
|---|---|---|---|---|---|
| Tillage | Residue | Fertilizer | Soluble-P (mg kg$^{-1}$) | NaHCO$_3$-Pi (mg kg$^{-1}$) | NaHCO$_3$-Po (mg kg$^{-1}$) |
| CT | HR | N0 | 3.3±0.01g | 44.9±0.25b | 18.2±0.45bd |
| | | N60 | 4.3±0.01f | 43.2±0.02c | 13.0±0.45ce |
| | | N80 | 5.5±0.01e | 35.9±0.01d | 15.6±0.78be |
| | | N100 | 13.4±0.00a | 35.0±0.02d | 15.6±0.78be |
| | | N120 | 11.2±0.01b | 25.4±0.01f | 13.0±0.45ce |
| | | N140 | 8.0±0.05c | 21.6±0.05h | 18.2±0.45bd |
| | LR | N0 | 2.4±0.02h | 24.2±0.14g | 15.6±0.00be |
| | | N60 | 3.3±0.02g | 21.6±0.05h | 15.6±0.78bce |
| | | N80 | 4.2±0.02f | 2.4±0.05p | 15.6±0.00be |
| | | N100 | 1.1±0.00k | 2.7±0.03op | 20.8±0.90bc |
| | | N120 | 1.0±0.01k | 4.8±0.00n | 15.6±0.00be |
| | | N140 | 1.6±0.03ij | 3.9±0.01no | 18.2±0.45bd |
| ST | HR | N0 | 5.5±0.01e | 46.0±0.03a | 18.2±0.45bd |
| | | N60 | 11.2±0.01b | 30.3±0.02e | 30.0±0.00ab |
| | | N80 | 11.2±0.01b | 14.4±0.02j | 20.8±0.90bc |
| | | N100 | 7.9±0.01 cd | 13.2±0.03k | 36.0±0.00a |
| | | N120 | 7.5±0.06d | 10.5±0.03l | 30.0±0.90ab |
| | | N140 | 5.7±0.03e | 10.0±0.03l | 28.0±0.78bc |
| | LR | N0 | 3.3±0.01g | 17.1±0.13i | 18.2±0.45bd |
| | | N60 | 3.2±0.00g | 10.8±0.03l | 20.8±0.45bc |
| | | N80 | 1.4±0.03ik | 9.8±0.01l | 23.4±0.00b |
| | | N100 | 1.7±0.01ij | 6.7±0.01m | 20.8±0.45bc |
| | | N120 | 1.7±0.01i | 4.6±0.02n | 10.4±0.45de |
| | | N140 | 1.2±0.07jk | 2.9±0.01op | 7.8±0.00e |
| LSD (p<0.05) | | | 0.04 | 0.11 | 0.08 |
| Level of Significance | | | | | |
| Tillage x Residue | | | ** | *** | * |
| Fertilizer x Tillage | | | *** | *** | ** |
| Fertilizer x Residue | | | *** | *** | *** |
| Fertilizer x Tillage x Residue | | | *** | *** | *** |

CT = conventional tillage; ST = strip tillage; HR = higher residue: 30 cm of rice and wheat residue + 100% mungbean residue, by height; LR = lower residue: 15 cm of rice and wheat residue + 100% mungbean residue; N0 = 0% of recommended dose; N60 = 60% of recommended dose; N80: 80% of recommended dose; N100: 100% of recommended dose; N120: 120% of recommended dose; N140: 140% of recommended dose.

ns, *, **, ***indicate significant difference at the level P > 0.05, P < 0.05, P < 0.01, P < 0.001, respectively. Means separated by same lower-case letter (a, b, and c) under each column were not significantly different at p > 0.05 among different treatment combinations.

Nitrogen fertilization significantly enhanced NaOH-Po buildup and the highest NaOH-Po (54.6 mg kg$^{-1}$) was observed under ST-HR-N120 and ST-HR-N140, highlighting the role of higher N rates.

## Effects of management practices on non-labile phosphorus fractions

The interaction between tillage, residue level, and N fertilizers significantly influenced the acid-P fractions in soil (P < 0.001, Table 3). The maximum value was observed in ST-HR-N60 (24.2 mg kg$^{-1}$), while the lowest occurred in ST-LR-N120 (2.0 mgkg$^{-1}$), indicating that moderate N levels under HR and ST enhance acid-P accumulation.

**Table 2. Moderately labile-P fractions in soil (depth: 0-15 cm) under long term conventional (CT) and strip (ST) tillage systems; high residue (HR) and low residue (LR) retention; and six N application rates.**

| Treatments | | | Moderately Labile Phosphorus Fractions | |
|---|---|---|---|---|
| Tillage | Residue | Fertilizer | NaOH-Pi (mg kg$^{-1}$) | NaOH-Po (mg kg$^{-1}$) |
| CT | HR | N0 | 5.6±0.01r | 20.8±0.45 cg |
| | | N60 | 8.6±0.02p | 18.2±0.45dh |
| | | N80 | 9.7±0.01o | 20.8±0.90 cg |
| | | N100 | 10.4±0.03o | 28.6±0.90bd |
| | | N120 | 13.4±0.00m | 33.8±0.90b |
| | | N140 | 16.6±0.03k | 33.8±0.90b |
| | LR | N0 | 7.2±0.05q | 15.6±0.78eh |
| | | N60 | 5.7±0.01r | 11.7±0.39fh |
| | | N80 | 7.7±0.10pq | 7.8±0.00h |
| | | N100 | 17.8±0.05j | 10.4±0.45gh |
| | | N120 | 18.2±0.00j | 10.4±0.45gh |
| | | N140 | 15.2±0.02l | 10.4±0.45gh |
| ST | HR | N0 | 22.3±0.03g | 15.6±0.78eh |
| | | N60 | 26.9±0.19e | 18.2±0.45dh |
| | | N80 | 33.2±0.09d | 31.2±0.78bc |
| | | N100 | 35.4±0.04c | 52.0±0.45a |
| | | N120 | 36.6±0.03b | 54.6±0.78a |
| | | N140 | 38.2±0.04a | 54.6±0.78a |
| | LR | N0 | 6.1±0.05r | 18.2±0.45dh |
| | | N60 | 12.4±0.01n | 17.1±0.25dh |
| | | N80 | 13.8±0.01m | 20.8±0.90 cg |
| | | N100 | 19.1±0.04i | 23.4±1.35bf |
| | | N120 | 21.1±0.01h | 23.4±1.35bf |
| | | N140 | 24.2±0.03f | 26.0±1.19be |
| LSD (p<0.05) | | | 0.09 | 1.28 |
| Level of Significance | | | | |
| Tillage x Residue | | | *** | ns |
| Fertilizer x Tillage | | | *** | ** |
| Fertilizer x Residue | | | *** | *** |
| Fertilizer x Tillage x Residue | | | *** | ns |

CT = conventional tillage; ST = strip tillage; HR = higher residue: 30 cm of rice and wheat residue + 100% mungbean residue, by height; LR = lower residue: 15 cm of rice and wheat residue + 100% mungbean residue; N0 = 0% of recommended dose; N60 = 60% of recommended dose; N80: 80% of recommended dose; N100: 100% of recommended dose; N120: 120% of recommended dose; N140: 140% of recommended dose.

ns, *, **, ***indicate significant difference at the level P>0.05, P<0.05, P<0.01, P<0.001, respectively. Means separated by same lower-case letter (a, b, and c) under each column were not significantly different at p>0.05 among different treatment combinations.

Residue-P showed a broader range (3.3–34.2 mg kg$^{-1}$) across all the treatment combinations and was significantly affected by the interaction of tillage, residue levels and N fertilizers rate (P<0.01, Table 3). The highest content (34.2 mg kg$^{-1}$) was found in ST-LR-N0, while the lowest (3.3 mg kg$^{-1}$) observed under ST-LR-N140. Residue-P decreased sharply with increasing N rates, particularly under LR treatments and tillage treatments didn't exhibit notable impacts on the residue-P fractions.

**Table 3. Non labile-P fractions in soil (depth: 0-15 cm) under long term conventional (CT) and strip (ST) tillage systems; high residue (HR) and low residue (LR) retention; and six N application rates.**

| Treatments | | | Non-labile Phosphorus Fractions | |
| --- | --- | --- | --- | --- |
| Tillage | Residue | Fertilizer | Acid-P (mg kg$^{-1}$) | Residue-P (mg kg$^{-1}$) |
| CT | HR | N0 | 7.0±0.09hi | 13.3±0.14bf |
| | | N60 | 10.7±0.12fh | 15.8±0.29bd |
| | | N80 | 10.0±0.00gh | 12.5±0.43cf |
| | | N100 | 14.5±0., e.g., | 15.6±0.000be |
| | | N120 | 8.1±0.08hi | 19.2±0.38b |
| | | N140 | 10.0±0.00gh | 9.7±0., e.g., |
| | LR | N0 | 7.5±0.00hi | 15.8±0.29bd |
| | | N60 | 14.5±0., e.g., | 15.8±0.29bd |
| | | N80 | 10.5±0.15fh | 15.9±0.30bd |
| | | N100 | 16.2±0.03be | 11.6±0.06 dg |
| | | N120 | 15.0±0.15df | 17.5±0.01bc |
| | | N140 | 24.0±1.31a | 12.4±0.03cf |
| ST | HR | N0 | 21.0±0.26ab | 19.2±0.14b |
| | | N60 | 24.2±0.14a | 14.2±1.15bf |
| | | N80 | 20.0±0.00ac | 9.2±0.29fh |
| | | N100 | 18.4±0.04be | 11.7±0.38 cg |
| | | N120 | 15.5±0.39ce | 15.8±1.04bd |
| | | N140 | 20.5±0.09ab | 5.8±0.14gh |
| | LR | N0 | 20.0±0.02ac | 34.2±0.14a |
| | | N60 | 19.4±0.05ad | 33.3±0.38a |
| | | N80 | 19.1±0.04be | 28.3±0.38a |
| | | N100 | 19.9±0.03ac | 12.5±0.22cf |
| | | N120 | 2.0±0.03j | 17.5±0.01bc |
| | | N140 | 3.7±0.05ij | 3.3±0.03h |
| LSD (p<0.05) | | | 0.48 | 0.6 |
| Level of Significance | | | | |
| Tillage x Residue | | | *** | * |
| Fertilizer x Tillage | | | *** | *** |
| Fertilizer x Residue | | | ns | *** |
| Fertilizer x Tillage x Residue | | | *** | ** |

CT = conventional tillage; ST = strip tillage; HR = higher residue: 30 cm of rice and wheat residue + 100% mungbean residue, by height; LR = lower residue: 15 cm of rice and wheat residue + 100% mungbean residue; N0 = 0% of recommended dose; N60 = 60% of recommended dose; N80: 80% of recommended dose; N100: 100% of recommended dose; N120: 120% of recommended dose; N140: 140% of recommended dose.

ns, *, **, ***indicate significant difference at the level P > 0.05, P < 0.05, P < 0.01, P < 0.001, respectively. Means separated by same lower-case letter (a, b, and c) under each column were not significantly different at p > 0.05 among different treatment combinations.

## Discussions

### Labile phosphorus fractions under different management practices

Labile phosphorus fractions, particularly soluble-P, NaHCO$_3$-Pi (inorganic), and NaHCO$_3$-Po (organic) are considered the most plant-available P forms [42] and act as the first responders to nutrient interventions. Our results showed that strip tillage (ST) combined with high residue (HR) retention significantly enhanced labile P fractions compared to conventional tillage (CT) with low residue (LR) system. This effect was especially pronounced under moderate nitrogen (N) application

rates (≤N100), suggesting a synergistic relationship among reduced soil disturbance, organic matter input, and optimized N supply. Our results are also in line with Sharma et al. [43] who indicated that individual tillage treatments did not significantly alter the soluble P content. The positive effect of HR on soluble-P contents supports previous findings [5,44–46] which observed that surface-retained residues reduce P adsorption by mineral surfaces due to increased soil organic matter and competitive anion interactions. According to Hou et al. [47], soluble-P may rise as a result of dissolved soil organic matter or organic acids buildup, which mobilize P through ligand exchange, restrict P sorption and enhance P solubility in soil. Our results align with findings from Reddy et al. [6], Kumawat et al. [48] and Redel et al. [49], who observed increased labile P accumulation in surface soils under conservation agriculture (CA) practices. The improvement under ST-HR may be attributed to minimal soil disturbance, reduced fertilizer-soil contact and less P fixation and increased soil organic carbon (SOC) improving microbial activity and promoting mineralization of organic P ($NaHCO_3$-Po) [45,50].

Higher N rates (≥N100) tended to decrease $NaHCO_3$-Po levels across tillage and residue treatments, indicating a possible shift from organic to inorganic P pools or increased mineralization under high N loading, or likely due to microbial turnover and mineralization rates exceeding organic inputs. The ST-HR-N0 treatment yielded the maximum $NaHCO_3$-Pi contents indicating that low N input under ST-HR conditions preserves labile Pi, possibly by limiting microbial immobilization or P uptake by crops. Under ST-LR and CT-LR, higher N reduced labile-Pi contents in soil. The drop in soil $NaHCO_3$-Pi concentration may be attributed to the increased N levels which promoted plant development and root P uptake and was subsequently distributed to other plant parts [51]. This depletion could also reflect intensified biological uptake, microbial immobilization, or conversion into less available forms due to acidification and fixation, a phenomenon previously noted by Zhao et al. [51] and Bolo et al. [52]. Although higher N rates (N120 and N140) can increase total biomass and potentially P uptake, they reduce labile-P pools, suggesting a mismatch between nutrient input and plant/microbial demand. According to Hu et al. [53], higher N rate promotes potential net N mineralization and nitrification processes which may raise the biotic P requirement [54]. Higher N rates (>N100) reduced labile P levels, potentially due to acidification effects of ammonium-based fertilizers, enhanced plant uptake, or microbial immobilization of P [51,53]. Moreover, ammonium ions ($NH_4^+$) from urea fertilizers can interact with soil particles and increase adsorption of labile-P fractions due to electrostatic interactions. Past researchers have argued that the soluble-P may be decreased as a result of P fixation, leaching and low phosphate ion mobility [12,52], because higher soluble-P is the result of a synergistic impact between increased accessible Pi and increased N [44]. These findings underscore the importance of synchronizing nutrient application with biologically driven P cycling under CA and also imply that optimizing N rate under CA can maintain labile P while minimizing environmental risks like eutrophication, groundwater toxicity etc.

## Moderately labile phosphorus fractions under different management practices

Moderately labile P (NaOH-Pi and NaOH-Po) which is chemisorbed to Fe/Al oxides or complexed with humic substances, appears to act as a transitional pool between labile and stable forms, contributing to long-term P buffering [11,55]. Conservation agriculture can improve the moderately labile P fractions [45] which as evident in the present study. Strip tillage with high residue and elevated N rates (≥ N100) consistently enhanced moderately labile P pools (NaOH-Pi and NaOH-Po), indicating that these forms are strongly influenced by biologically mediated processes and soil chemical conditions. Moreover, our study showed that NaOH-P fractions were consistently higher under ST than CT, emphasizing the role of reduced disturbance in conserving organic inputs and promoting microbial processes. The concurrent increase in NaOH-Po suggests enhanced microbial biomass turnover and organic matter accumulation, especially under HR conditions.

High nitrate ($NO_3^-$) concentrations released from N fertilizers might acidify soils, enhancing desorption of Fe/Al-bound P and shifting equilibrium toward NaOH-Pi fractions [56,57]. Moreover, the reasons might be soil pH changes, microbial dynamics, ionic interactions, and redox conditions that are influenced by increased $NO_3^-$–N concentrations. Higher $NO_3^-$ ions could enhance microbial activity and alter the adsorption-desorption equilibrium that convert organic phosphorus

to inorganic forms. The findings of our study align with other researchers [56–59], who found that N-induced microbial activity facilitates the mobilization of moderately labile P pools. Higher NaOH-Pi pools with higher N rates in our study can be attributed to mobilization of NaOH-Pi fraction to augment the sufficient amount of accessible Pi for crop growth. The decline in NaOH-Pi pools indicated that the size of the NaHCO$_3$-Pi pool in the soil might not be sufficient to meet crop P needs [60] and thus NaOH-Pi pools were converted to labile Pi fractions. The NaOH-Po pool may serve as the primary source of plant-available P in the soil and sustain plant-available P levels [61]. Lower mineralization would be expected at lower N supply resulting in lower NaOH-Po pools. In situations where soil P supply is low, Po mineralization serves as crops' primary source of available Pi resulting in lower NaOH-Po pools [62]. Thus, adoption of optimum N rate is crucial for preventing loss of moderately labile-P pools through erosion, runoff or other factors.

This study also contributes novel insight into how management-driven shifts in microbial-mediated processes affect intermediate P pools. It also emphasizes the buffering role of NaOH-P fractions, suggesting that regulating N inputs with CA practices can sustain crop-available P over time while minimizing external P input requirements. Nevertheless, this study did not directly measure microbial biomass or enzyme activity, which limits our ability to conclusively link biological processes with P transformations. Future research should integrate microbial and biochemical indicators to strengthen the understanding.

## Non-labile phosphorus fractions under different management practices

The recalcitrant P pools (acid-P and residual-P) are generally considered stable, non-labile and unavailable for immediate plant uptake but represent important long-term nutrient reserves. The acid-P fraction contained P bonded to Ca [55] and the quantity is high in alkaline soil due to the high stability of Ca-P [44]. The levels of acid-P varied depending on the tillage and residue management coupled with N rate. In our study, Acid-P levels increased under ST and HR that was opposite to the findings of Haokip et al. [45] who found no significant interaction between tillage and nutrients on acid-P. Acid-P was higher under ST, particularly when coupled with lower N rate (≤N100) and HR. In soils near neutral pH (as in our site), Ca-bound P is not readily soluble; however, high N may contribute to acidification and enhance solubilization, as suggested by Jing et al. [63]. This may explain the decline in acid-P at higher N rates (N120, N140), possibly due to hydrolysis or transformation of a relatively resistant acid-P pool into labile or moderately labile forms under increased plant/microbial demand [63,64]. The acid-P fractions' altered concentrations suggest that these fractions may be involved in long-term P cycling. Under CA, the acid-P fractions may function as a slow-release storage of P, ensuring that plants constantly have a supply of available P even when short-term inputs are altered, supporting long-term soil fertility and crop production. Since crop demand for P absorption could be increased by high N fertilizer, it is possible that acid-P depletion occurred in addition to labile Pi fractions [64]. A portion of the acid-P pool's hydrolysis products would move straight into the NaHCO$_3$-Pi pool. Nevertheless, the portions of Pi near the surface of the original mineral particles might easily be re-adsorbed and fixed again and merged into acid-P if the released P or the immediately generated NaHCO$_3$-Pi cannot be utilized by crops or microorganisms in time; and it is also possible for other P pools to partially complement acid-P pools [63]. Through better soil, crop and nutrient management, these non-labile P pools can be utilized over the multiple cropping seasons and chemical P inputs can be minimized.

The residual-P pools are unlikely to provide short-term contributions to the soil solution and plant nutrients [65]. Residual-P, consisting of mineral-occluded and highly refractory forms, declined under high N rates, particularly under ST with LR. This pattern may reflect increased P demand, release of P from residual P pools and mobilization via microbial activity, as supported by findings of Fan et al. [46] and Mahmood et al. [58]. Lower residues might have boosted residual-P at lower N rates, whereas higher residues might have decreased its mobilization due to the residual-P's persistence, resistance, and insoluble nature. The gradual decrease in residual P suggests that even recalcitrant pools can contribute to P cycling under prolonged cultivation and appropriate stimulation, reinforcing the importance of integrated nutrient management [65]. Interestingly, residual-P content declined with increasing N rate and HR indicating that residual-P can

be biologically mobilized over time, especially under improved soil conditions and active root growth [66]. An increase in biomass among the plants (particularly root length and root hair density) during P shortage can enhance plant performance and greatly boost residual P absorption [66,67]. When agricultural residues are added to soil systems, the residual P portion could function as a sink [68]. The transformation of residual P into more available forms under CA practices and moderate N rates suggests that such systems can gradually tap into native soil reserves, reducing dependence on external P inputs. However, the variability in non-labile P response across treatments and the lack of significant trends for some interactions highlight the complexity of these pools and their resistance to short-term changes. It also suggests the need for longer monitoring or inclusion of isotopic P labeling techniques to track P transformations.

## Conclusions

This long-term field study demonstrates that phosphorus availability and transformation in intensively managed cropping system are significantly influenced by the interactive effects of tillage practices, crop residue retention, and nitrogen input levels. Specifically, strip tillage combined with high residue retention and moderate to higher nitrogen application enhanced labile and moderately labile phosphorus pools, while conventional tillage with lower residue incorporation under lower nitrogen rates tended to deplete these pools and increase phosphorus fixation into non-labile forms. A larger amount of non-labile phosphorus pool under both conventional and conservation agricultural practices was obtained with moderate nitrogen inputs, however, higher rates of nitrogen fertilization reduced this pool. These findings underscore the mechanistic link between soil tillage and nitrogen management and the maintenance of dynamic phosphorus pools that is critical for the sustenance of soil fertility and sustainable crop productivity. The novelty of this study lies in its comprehensive evaluation of phosphorus fractions under long-term, real-world agronomic practices, moving beyond total phosphorus to show how specific management choices alter available and stable phosphorus pools. The study contributes novel evidence on phosphorus dynamics under tropical conservation agriculture systems and provides valuable insights for refining site-specific efficient nutrient management strategies, particularly in intensive cereal-based cropping systems of South Asia. The findings provide a scientific basis for promoting conservation agriculture practices that enhance phosphorus use efficiency and contribute to long-term soil fertility. However, the study was conducted under specific agro-climatic and soil conditions, which may limit the direct applicability of results to other regions. Future studies should explore microbial and enzymatic processes associated with phosphorus cycling and assess crop uptake dynamics in conjunction with soil phosphorus fractions.

## Supporting information

**S1 Table. Supporting information_raw dataset.**
(XLSX)

## Acknowledgments

The authors gratefully acknowledge the logistic support from both the Department of Soil Science at Bangladesh Agricultural University and Khulna Agricultural University.

## Author contributions

**Conceptualization:** Md. Mofizur Rahman Jahangir.

**Data curation:** Sunjana Akter, Afsana Mimi Eiti Mony, Md. Mofizur Rahman Jahangir.

**Formal analysis:** Nusrat Jahan Mumu.

**Investigation:** Nusrat Jahan Mumu, Md. Mofizur Rahman Jahangir.

**Methodology:** Nusrat Jahan Mumu.

**Project administration:** Md. Mofizur Rahman Jahangir.

**Resources:** Nusrat Jahan Mumu, Sunjana Akter, Afsana Mimi Eiti Mony, Md. Mofizur Rahman Jahangir.

**Software:** Sunjana Akter, Afsana Mimi Eiti Mony.

**Supervision:** Md. Mofizur Rahman Jahangir.

**Validation:** Md. Mofizur Rahman Jahangir.

**Visualization:** Md. Mofizur Rahman Jahangir.

**Writing – original draft:** Nusrat Jahan Mumu.

**Writing – review & editing:** Sunjana Akter, Afsana Mimi Eiti Mony, Jannatul Ferdous, Nushaiba Atiq Taima, Most. Khatiza Khatun, Md. Mofizur Rahman Jahangir.

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
