## [Decision Letter · Decision Letter 0]

24 Apr 2025

PONE-D-25-17789Effect of long-term conservation agriculture with nitrogen rates on soil available and labile phosphorus poolsPLOS ONE

Dear Dr. Nusrat Jahan Mumu,

Thank you for submitting your manuscript to PLOS ONE. After careful consideration, we feel that it has merit but does not fully meet PLOS ONE’s publication criteria as it currently stands. Therefore, we invite you to submit a revised version of the manuscript that addresses the points raised during the review process.

We look forward to receiving your revised manuscript.

Kind regards,

Abhay Omprakash Shirale, PhD

Academic Editor

PLOS ONE

Reviewers' comments:

Reviewer's Responses to Questions

**Comments to the Author**

1. Is the manuscript technically sound, and do the data support the conclusions?

Reviewer #1: Yes

Reviewer #2: Partly

Reviewer #3: Yes

2. Has the statistical analysis been performed appropriately and rigorously? 

Reviewer #1: Yes

Reviewer #2: Yes

Reviewer #3: Yes

3. Have the authors made all data underlying the findings in their manuscript fully available?

Reviewer #1: Yes

Reviewer #2: No

Reviewer #3: Yes

4. Is the manuscript presented in an intelligible fashion and written in standard English?

Reviewer #1: Yes

Reviewer #2: No

Reviewer #3: No

5. Review Comments to the Author

Reviewer #1: The manuscript presents a long-term field study investigating how conservation agriculture (CA) practices, particularly tillage and crop residue management, influence soil phosphorus (P) fractions under varying nitrogen (N) rates. This is a timely and relevant topic, particularly in regions facing nutrient limitations and environmental sustainability challenges. The manuscript is generally well-organized and contributes useful findings to the literature on nutrient dynamics under CA. There are some comments, which are to be incorporated in order to improve the manuscript, as given below:

Introduction

*Hypothesis of the work is not well formulated in the ‘Introduction’ section. The authors did not present a novel justification for carrying out this study. What is the hypothesis of the present study?

*It is insufficient and needs more improvement.

*The novelty of the work must be identified and stated more carefully. The authors have to try to explain why this paper is relevant to the wider readership.

*Authors should show the limitations of previous papers.

Results

*Result should be written in concise way.

*Quality of figures should be improved.

Discussion

*Need more improvement. It is too superficial and not a meaningful discussion. Author should try to strengthen the discussion part.

*Put reason why this type of result is obtained.

Conclusions

*Authors need to rephrase the "Conclusions" section.

*Add some limitations, underscore the scientific value added of your paper, and/or the applicability of your findings/results and future scope of study.

*Overall, the manuscript needs extensive language editing and formatting. There are so many typography mistakes in the manuscript.

Reviewer #2: Comments to authors:

Effect of long-term conservation agriculture with nitrogen rates on soil available and labile phosphorus pools

Manuscript Number: PONE-D-25-17789

Detail suggestions are given below:

1. Title: What type of “effect” should be specified? This is journal article not thesis.

2. Key words: Title words should not be kewords.

3. Abstract: Hypothesis is not clear and results should be measurable (% increase or decrease). Higher N rates did not reduce all P fractions.

4. Title words should not be key words.

5. Introduction: There are too many irrelevant topics included here. There are huge references on. But your references are too old, use latest references (Some link attached here ; https://doi.org/10.1111/sum.13153;
https://doi.org/10.1007/s42729-024-02009-z;
https://doi.org/10.1007/s10705-024-10348-7;
https://doi.org/10.1016/j.still.2022.105474;
http://dx.doi.org/10.1007/s10705-023-10261-5;
https://sssb1958.com/wp-content/uploads/2024/12/U-Kumar.pdf). Lack of novelty and research scope. Introduction should be rearranged by the sequence of - important of the research, what is the known about the topic, unknown about the topic, why this research is important and finally aims of this research.

Line 112: AEZ 9 is Old Brahmaputra Floodplain not Old Himalayan Piedmont Plain.

6. Materials and methods: Line 144: Medium Highland Sonatala soil series is poorly drained soil not moderately drained.

Line 151-152: What are the crops diversifications? Provide the 12 years cropping pattern.

Line 162: RD? Elaborate first.

Line 168-171: Fertilizers doses for each crop are not clear. Provide in a table and give it to supplementary data.

Line 180-182: 0-15 soil depth? Sample collection procedure should be clear specially core sampler.

Line 186: descried by 29.?

7. Need to rewrite the results section. Results are not described appropriately. Several results are confusingly and repeatedly presented. If interaction is significant, present only interaction effect results in the table or figures. No need to describe main factor results. Result should be measurable e.g., how much increase or decrease must be specified. In general, several paragraphs are very repetitive of the information provided in Tables. It is recommended to highlight more important results, rather than repeat results presented in tables. Different P fractions have interlink with other soil properties like bulk density, texture, soil organic carbon, N, and extractable K, S, Zn, B, Ca, Mg, Fe, Mn, Cu etc. So, need to describe what is the relation among them? Crops yield is also need to describe to understand the optimum production with P fractions. The manuscript results has lack of novelty and unable to give specific message to the scientific communities.

8. Discussion is not another result. The discussion should link all of your findings. Each paragraph should be well structured and interconnected (“telling a story”). For example, in each paragraph of the discussion section, authors should begin with a sentence that introduces what the paragraph is about of their findings, link their findings with those in the literature, and finish with what are the main points ('take-home-message'). Authors also need to discuss about the limitations and lackings of their study and its justifications. Please, a deeper scientific interpretation of your findings in the discussion section is strongly suggested. The novelty and implication of your study should also be highlighted throughout this section. In addition, I suggest using more actualized.

9. The conclusions section should not be a summary of your study or an extension of the discussion or results. This section should illustrate the mechanistic links of findings obtained under applied treatments. The conclusions should answer the hypothesis of your study and should focus on the implication of your findings. Please, avoid using abbreviations and acronyms in this section. Remember that the conclusions must be self-explanatory. This section should highlight the novelty and implication of your study also.

10. Avoid unnecessary tables and figures and they must be self-explanatory.

11. English improvement is needed in whole MS

The purpose, novelty and originality of your study seem to be unclear and incomplete throughout the manuscript. The manuscript is unable to give new message to the scientific community. It is not acceptable in the present form. It requires substantial improvement in your manuscript.

Reviewer #3: Reviewer Comments:

While the information presented in the manuscript is valuable and merits publication, the manuscript requires substantial revision to enhance its clarity, coherence, and overall quality.

Abstract:

The abstract should provide a concise yet comprehensive summary of the entire manuscript. It must include brief statements covering the background, objectives, methodology, key findings, and major conclusions. Importantly, the abstract should be focused and well-structured, offering the reader a clear snapshot of the study's purpose and outcomes.

Introduction:

The introductory section needs to be significantly strengthened. It should begin with a focused discussion on phosphorus (P) chemistry and its dynamics in soil systems. Following this, the authors should clearly articulate how agronomic management practices—particularly varying tillage methods, amount of residue retention, and nitrogen (N) fertilization rates—affect soil P dynamics and availability, ultimately influencing crop growth and productivity. Establishing this context is essential to justify the research objectives and the significance of the study.

Results and Discussion:

The most critical findings should be highlighted in this section. Special attention should be given to explaining the individual and interactive effects of tillage practices, crop residue levels, and N fertilization on phosphorus behavior and crop performance. If statistical interactions are observed, these must be explained clearly and meaningfully. The authors should interpret why and how such interactions occurred, grounded in sound agronomic or biochemical reasoning.

Conclusion:

The conclusion should be succinctly focused. It must restate the main findings of the study and their practical implications. Avoid reiterating background information or including new data. Instead, the conclusion should draw attention to the most relevant and impactful results and how they advance current understanding or practices.

Tables and Data Presentation:

All data presented in tables must be systematically organized according to the experimental design, particularly reflecting the split-plot arrangement. Clear labeling, consistent formatting, and appropriate statistical indicators (e.g., LSD, SE, p-values) are essential to improve readability and interpretation.

General Recommendations:

The authors are strongly encouraged to carefully address the specific comments and suggestions embedded within the manuscript text. These comments are intended to enhance the overall structure, language quality, and scientific rigor of the paper. Ensuring logical flow and technical accuracy throughout will significantly improve the manuscript’s suitability for publication.

6. PLOS authors have the option to publish the peer review history of their article (what does this mean? ). If published, this will include your full peer review and any attached files.

**Do you want your identity to be public for this peer review?** For information about this choice, including consent withdrawal, please see our Privacy Policy .

Reviewer #1: No

Reviewer #2: **Yes: ** Utpol Kumar

Reviewer #3: **Yes: ** Rafiq Islam

---

## [Author Response · Author response to Decision Letter 1]

1 Jul 2025

Dear Dr. Abhay Omprakash Shirale,

Thank you for giving us the opportunity to submit the revised manuscript entitled “Effect of long-term conservation agriculture with nitrogen rates on soil available and labile phosphorus pools” for publication in PLOS ONE. We appreciate the considerable time and efforts you and the reviewers put into providing suggestions on our manuscript. We incorporated the reviewers' suggestions in our manuscript.

Reviewer 1

1. Comment: Hypothesis of the work is not well formulated in the ‘Introduction’ section. The authors did not present a novel justification for carrying out this study. What is the hypothesis of the present study?

Author response: Thank you for pointing this out. we have added the hypothesis of the present study in the text and also a clear justification for this hypothesis.

2. Comment: Introduction is insufficient and needs more improvement. The novelty of the work must be identified and stated more carefully. The authors have to try to explain why this paper is relevant to the wider readership. Authors should show the limitations of previous papers.

Author response: CA system has been recognized to change the nutrient dynamics and stoichiometry but how these dynamics interact with N rate for P availability in floodplain soils in subtropical agroecosystems are limited. We added the limitations of previous research works and also added the novelty of our research in the text.

3. Comment: Result should be written in concise way.

Author response: We appreciate the reviewer’s feedback. The result section has been rewritten following the valuable comments of the reviewers.

4. Comment: Quality of figures should be improved.

Author response: The quality of figure has been improved and a new one is added.

5. Comment: Author should try to strengthen the discussion part. Put reason why this type of result is obtained.

Author response: We have rewritten this section and improved the quality of the section with science-based arguments on the findings.

6. Comment: Authors need to rephrase the "Conclusions" section. Add some limitations, underscore the scientific value added of your paper, and/or the applicability of your findings/results and future scope of study.

Author response: We have added the suggested contents on conclusion section of the manuscript.

7. Comment: Overall, the manuscript needs extensive language editing and formatting. There are so many typography mistakes in the manuscript.

Author response: Thank you for this suggestion. The manuscript has been thoroughly edited by a native English speaker, Dr. Richard Bell, Professor at Murdoch University, Australia. The readability and English grammar and use of the article are now improved.

Reviewer 2

1. Comment: In title what type of “effect” should be specified?

Author response: Thank you for pointing this out. We have made a correction in the title.

2. Comment: Title words should not be keywords.

Author response: We have added new keywords different from title words.

3. Comment: Hypothesis is not clear and results should be measurable (% increase or decrease) in abstract. Higher N rates did not reduce all P fractions.

Author response: Abstract section has been updated following the comments of reviewers.

4. Comment: There are too many irrelevant topics included in introduction section. There are huge references on. But your references are too old. Lack of novelty and research scope. Introduction should be rearranged by the sequence of - important of the research, what is the known about the topic, unknown about the topic, why this research is important and finally aims of this research.

Author response: We rewritten the section by rearranging the sequence, adding some new information following the reviewer’s suggestions.

5. Comment: Line 112: AEZ 9 is Old Brahmaputra Floodplain not Old Himalayan Piedmont Plain. Line 144: Medium Highland Sonatala soil series is poorly drained soil not moderately drained.

Author response: Thank you for pointing these out. We have now made the necessary corrections in the manuscript.

6. Comment: Line 151-152: What are the crops diversifications? Provide the 12 years cropping pattern.

Author response: Thank you for your feedback. Although we agree that this is an important consideration, we didn’t provide 12yrs cropping pattern in details in this manuscript because it was already mentioned that a wheat–mungbean–rice cropping pattern was continued at our experimental location for 12 years with the same management to understand the long-term effects of CA on soil biogeochemistry and crop productivity.

7. Comment from Reviewer 2: Line 162: RD? Elaborate first.

Author response: We elaborated the term in the manuscript.

8. Comment: 168-171: Fertilizers doses for each crop are not clear. Provide in a table and give it to supplementary data.

Author response: While we appreciate the reviewer’s feedback, but the given reference has explained the crop management in a great detail.

9. Comment: 180-182: 0-15 soil depth? Sample collection procedure should be clear specially core sampler.

Author response: We revised the methodology and mentioned that the samples were collected with the help of an auger at 0-15 cm soil depth.

10. Comment: Need to rewrite the results section.

Author response: The result section has been rewritten following the valuable comments of the reviewers.

11. Comment: Discussion is not another result. The discussion should link all of your findings. Each paragraph should be well structured and interconnected.

Author response: We have rewritten this section and tried to improve the quality and strengthen the arguments for the obtained results.

12. Comment: conclusions must be self-explanatory. This section should highlight the novelty and implication of your study also.

Author response: The conclusion section has been rewritten following the valuable comments of the reviewers.

13. Comment: Avoid unnecessary tables and figures and they must be self-explanatory.

Author response: We think the figures makes a valuable contribution to the methodology section. But we have removed some of the tables that are unnecessary and made some corrections to improve the readability and visibility of data.

14. Comment: English improvement is needed in whole MS.

Author response: The manuscript has been edited by a native English speaker, Dr. Richard Bell, Professor, Murdoch University, Australia.

Reviewer 3

1. Comment: The abstract should be focused and well-structured, offering the reader a clear snapshot of the study's purpose and outcomes.

Author response: Thank you for the valuable suggestion. The abstract has been improved now.

2. Comment: The introductory section needs to be significantly strengthened. It should begin with a focused discussion on phosphorus (P) chemistry and its dynamics in soil systems. Following this, the authors should clearly articulate how agronomic management practices—particularly varying tillage methods, amount of residue retention, and nitrogen (N) fertilization rates—affect soil P dynamics and availability, ultimately influencing crop growth and productivity. Establishing this context is essential to justify the research objectives and the significance of the study.

Author response: Thank you for this suggestion. We revised the manuscript and corrected the introduction section according to reviewer’s valuable comments.

3. Comment: The most critical findings should be highlighted in result section. Special attention should be given to explaining the individual and interactive effects of tillage practices, crop residue levels, and N fertilization on phosphorus behavior and crop performance. If statistical interactions are observed, these must be explained clearly and meaningfully. The authors should interpret why and how such interactions occurred, grounded in sound agronomic or biochemical reasoning.

Author response: We appreciate the reviewer’s feedback. The result section has been rewritten following the valuable comments of the reviewers.

4. Comment: The conclusion should be succinctly focused. It must restate the main findings of the study and their practical implications. Avoid reiterating background information or including new data. Instead, the conclusion should draw attention to the most relevant and impactful results and how they advance current understanding or practices.

Author response: Thank you for the valuable suggestion. As suggested by the reviewer, we have tried to improve the conclusion.

5. Comment: All data presented in tables must be systematically organized according to the experimental design, particularly reflecting the split-plot arrangement. Clear labeling, consistent formatting, and appropriate statistical indicators (e.g., LSD, SE, p-values) are essential to improve readability and interpretation.

Author response: Thank you for pointing these out. We made the necessary corrections in Tables following the comments of reviewers.

---

## [Decision Letter · Decision Letter 1]

8 Sep 2025

Long-term conservation agriculture with optimum nitrogen fertilization improves soil phosphorus availability

PONE-D-25-17789R1

Dear Dr. Mumu,

We’re pleased to inform you that your manuscript has been judged scientifically suitable for publication and will be formally accepted for publication once it meets all outstanding technical requirements.

Kind regards,

Abhay Omprakash Shirale, PhD

Academic Editor

PLOS ONE

Additional Editor Comments (optional):

Reviewer #1:

Reviewer #3:

Reviewers' comments:

Reviewer's Responses to Questions

**Comments to the Author**

1. If the authors have adequately addressed your comments raised in a previous round of review and you feel that this manuscript is now acceptable for publication, you may indicate that here to bypass the “Comments to the Author” section, enter your conflict of interest statement in the “Confidential to Editor” section, and submit your "Accept" recommendation.

Reviewer #1: All comments have been addressed

Reviewer #3: All comments have been addressed

2. Is the manuscript technically sound, and do the data support the conclusions?

Reviewer #1: Yes

Reviewer #3: Yes

3. Has the statistical analysis been performed appropriately and rigorously? 

Reviewer #1: Yes

Reviewer #3: Yes

4. Have the authors made all data underlying the findings in their manuscript fully available?

Reviewer #1: Yes

Reviewer #3: Yes

5. Is the manuscript presented in an intelligible fashion and written in standard English?

Reviewer #1: Yes

Reviewer #3: Yes

6. Review Comments to the Author

Reviewer #1: The authors revised the paper in response to the suggestions. And the paper will be accepted for publication in its current form.

Reviewer #3: (No Response)

7. PLOS authors have the option to publish the peer review history of their article (what does this mean? ). If published, this will include your full peer review and any attached files.

**Do you want your identity to be public for this peer review?** For information about this choice, including consent withdrawal, please see our Privacy Policy .

Reviewer #1: No

Reviewer #3: **Yes: ** Rafiq Islam

---

## [Editor Report · Acceptance letter]

PONE-D-25-17789R1

PLOS ONE

Dear Dr. Jahan Mumu,

I'm pleased to inform you that your manuscript has been deemed suitable for publication in PLOS ONE. Congratulations! Your manuscript is now being handed over to our production team.

Kind regards,

on behalf of

Dr. Abhay Omprakash Shirale

Academic Editor

PLOS ONE